# More Numerically Accurate Algorithm for Stiff Matrix Exponential

**Teddy Lazebnik** [1,2,*] and **Svetlana Bunimovich-Mendrazitsky** [1]

1 Department of Mathematics, Ariel University, Ariel 4070000, Israel; svetlanabu@ariel.ac.il
2 Department of Cancer Biology, Cancer Institute, University College London, London WC1E 6BT, UK
* Correspondence: t.lazebnik@ucl.ac.uk

**Abstract:** In this paper, we propose a novel, highly accurate numerical algorithm for matrix exponentials (MEs). The algorithm is based on approximating Putzer's algorithm by analytically solving the ordinary differential equation (ODE)-based coefficients and approximating them. We show that the algorithm outperforms other ME algorithms for stiff matrices for several matrix sizes while keeping the computation and memory consumption asymptotically similar to these algorithms. In addition, we propose a numerical-error- and complexity-optimized decision tree model for efficient ME computation based on machine learning and genetic programming methods. We show that, while there is not one ME algorithm that outperforms the others, one can find a good algorithm for any given matrix according to its properties.

**Keywords:** decision tree for a numerical algorithm; stiff matrix exponential; Putzer approximation

**MSC:** 65L04; 65L08

## 1. Introduction and Related Work

A matrix exponential (ME) is a function defined as the solution of a system of *n* linear, homogeneous, first-order, ordinary differential equations (ODEs) with constant coefficients. Equation (1) outlines a formal writing of the ME function:

$$Y(t) = e^{Mt}Y(0) \leftrightarrow Y'(t) = MY(t), \ M \in \mathbb{C}^{n \times n}, \tag{1}$$

where $M \in \mathbb{C}^{n \times n}$ is an arbitrary matrix, $Y(t)$ is a dynamic over time, and $Y(0)$ is the initial condition of the dynamic system.

An ME is a widely used function [1–5]. For instance, it is used in linear control systems in order to find the state space, as the ME plays a fundamental role in the solution of the state equations [6,7]. In a more general sense, it plays a role in exploring and solving a wide range of dynamical systems represented by a set of ODEs [1]. Due to their usefulness in many applications, MEs have been widely investigated [1,8–11].

The analytical solution of Equation (1) takes the form [8]:

$$e^M = \sum_{i=0}^{\infty} \frac{M^i}{i!}. \tag{2}$$

Equation (2) is also known as the *naive* ME algorithm. It is numerically unstable, slow, and relatively inaccurate when considering floating-point arithmetic [1]. Recently, in the context of modern computer systems, numerical computation of MEs has become more relevant than ever before [1,12]. One of the main challenges associated with computing MEs is the numerical accuracy for any given matrix [9].

One can roughly divide the numerical ME algorithms into two groups: algorithms designed specifically to exploit a property of some groups of matrices (for example, diagonal

matrices) and algorithms that are suitable for all matrices. Usually, when one uses an algorithm from the first group, the obtained result is accurate and the computation is stable since the algorithm is explicitly designed to handle the given input matrix. If, for some reason, one uses an inappropriate matrix in such an algorithm, the algorithm more often than not produces a (highly) incorrect result. Algorithms from the second group are usually found in large computational systems as they do not require that the result of the computation, which takes place before the computation of the ME algorithm, satisfies a given condition. This general nature makes these algorithms usable in a wider context [1]. However, each algorithm in this group has a set of matrices that result in large errors or even divergence during computation [1].

One can name several popular examples of such algorithms. First, there is a Taylor series method with a stop condition that is based on Peano's remainder being smaller than some pre-defined threshold [13]. As a numerical method, it is often slow and inaccurate [1]. Specifically, matrices with small values, in absolute terms, are more likely to produce large errors due to cancellation errors. Second, the Pade-approximation-based algorithm is a two-parameter approximation of the ME Taylor series algorithm, which makes it more robust and well-performing after using scaling and squaring algorithm [1]. However, these algorithms obtain poor results, with the norm of the matrix $||M||$ being large. Third, the Cayley–Hamilton ME algorithm provides another form of approximation of the Taylor series. The coefficients of the decomposition of the input matrix are very sensitive to round-off errors as they generate a large error, specifically when the rank of the input matrix ($M$) is significantly smaller than that of the diminution of the matrix [8].

Another family of algorithms takes advantage of the eigenvalues of the input matrix. For instance, there are Lagrange-interpolation-based methods [8]. These methods provide very accurate results on average [1]. However, for an input matrix with close but not equal eigenvalues (i.e., $|\lambda_j - \lambda_k| << 1$), this method has a significant cancellation error as a result of dividing each iteration by $|\lambda_j - \lambda_k|$. Matrices with eigenvalues with great algebraic multiplicity will produce significant error and make the algorithm unstable. In a similar manner, the Newton interpolation algorithm is a good example as well. This algorithm is based on a recursive decomposition method of the input matrix and suffers from the same issues as the Lagrange interpolation algorithm.

An additional family of algorithms is the general-purpose ODE solvers. The main idea is that the ME is defined as a solution for a system of ODEs (see Equation (1)). Therefore, it is possible to solve Equation (1) numerically using different approaches such as the Euler, exponential integrator, and Runge–Kutta methods [14–16]. The Krylov methods are widely used to solve MEs for which the matrices are large and sparse [17]. These methods are widely adapted due to the fast and accurate results they produce for sparse matrices, which are becoming more common in multiple applications [17,18]. This class of methods is mainly dependent on the norm of the input matrix [19,20]. For cases for which the input matrix is negative definite, the condition number defines the convergence rate and error boundary [17,21]. However, this class of methods does not allow obtaining the ME itself but the product of the ME with a vector. While of interest in multiple applications, in this work, we aim to obtain the ME, as it can be both analyzed by itself and later multiplied with any vector easily.

Moreover, a large body of work investigates the numerical computation of MEs for the transient analysis of continuous-time Markov chains [22]. Multiple numerical algorithms for MEs are proposed: mostly based on the uniformization method [23]. These methods are shown to handle cases for which stiffness occurs. Nonetheless, [22] shows that even modern continuous-time Markov chain solvers are outperformed by Pade approximation combined with scaling and squaring. Recently, further works focused on the computation time and computational memory requirements of ME as matrices grow in size for multiple realistic cases [24,25].

In this study, we propose a novel L-EXPM algorithm that numerically solves MEs with a high level of accuracy. The L-EXPM is designed to tackle the issues of previous methods

by combining the advantages of series-based algorithms (for example, Pade) and the eigenvalue-based methods (for example, Newton) by approximating Putzer's algorithm.

The remainder of the paper is organized as follows. First, we introduce the proposed L-EXPM algorithm with asymptotic complexity and memory consumption analysis and prove it solves MEs. Second, we numerically evaluate the numerical accuracy of the proposed L-EXPM algorithm relative to other ME algorithms on stiff matrices. Third, we propose a complexity- and numerical-accuracy-optimized decision tree model for numerically solving ME. Finally, we conclude the usage of the proposed L-EXPM with a decision tree and suggest future work.

## 2. The L-EXPM Algorithm

### 2.1. Algorithm

Numerical computation of MEs using L-EXPM is aimed to reduce the error in computing MEs for any given complex square matrix. L-EXPM is based on Putzer's algorithm [26] for the decomposition of MEs. L-EXPM handles two steps of the original algorithm from a numerical perspective. First, it finds the eigenvalues of the input matrix ($M$). This is not a trivial task, especially for large-size matrices. However, recent power methods, such as the Lanczos algorithm [27], are able to obtain the needed eigenvalues with decent accuracy. Second, rather than solving a system of ODEs recursively to obtain the coefficients of the decomposed matrices, the coefficients are iteratively approximated via an analytical solution of the system of ODEs. A schematic view of the algorithm's structure is shown in Figure 1.

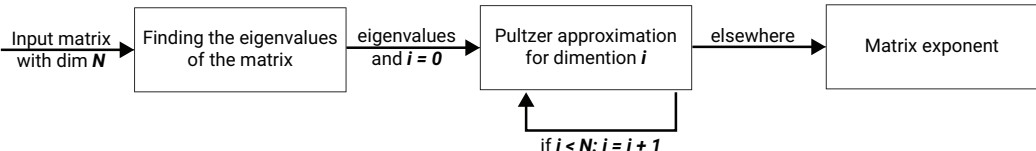

**Figure 1.** A schematic view of the algorithm's structure.

The L-EXPM algorithm takes a square complex (or real) matrix and returns the ME of this matrix. It works as follows: In line 2, the vector of eigenvalues of the input matrix is obtained using some algorithm that finds numerically the eigenvalues of the input matrix. Any algorithm may be used. Specifically, the Lanczos [27] algorithm has been used with the stability extension proposed by Ojalvo and Newman [28] and the spatial bisection algorithm [29]. In line 3, the eigenvalues that are too close to each other are converted to the same eigenvalue, and eigenvalues that are too close to 0 are converted to 0. In lines 4–5, the initial $r$ and $P$ (the input matrix decomposition ($P$) and its coefficient) from Putzer's algorithm are initialized. Lines 6–11 are the main loop of the algorithm, with line 8 being the sum of $r_{i+1}P_i$, while $r_i$ in line 9 is an approximation of the $r_i$ shown in [26], as shown in Theorem 1. Specifically, $c_l$ and $a_k$ are the coefficients of the polynomial–exponent representation of $r_i$, as later described in Equation (10). Line 10 is the iterative calculation of $P_i$.

In line 3, $controlEigs(M)$ is a function that replaces every two eigenvalues that satisfy $||\lambda_i - \lambda_j|| < \epsilon$ or one eigenvalue $||\lambda_i|| < \epsilon$, where $\epsilon$ is an arbitrary small threshold with the same eigenvalue $\lambda_i$ or 0.

We examine the L-EXPM algorithm's storage requirements and complexity (Algorithm 1). Analyzing the worst-case scenario, asymptotic complexity, and memory consumption can be performed by dividing the algorithm into two parts: lines 2–5 and 6–11. Assuming that the worst complexity and memory consumption of the algorithm that finds the eigenvalues of the input matrix are $O(E_c)$ and $O(E_m)$, respectively, and that the input matrix is $n$-dimensional, then it is easy to see that $controlEigs$ has $O(n^2)$ complexity and $O(n)$ memory consumption as it compares every two values in a vector. Lines 4 and 5 are the initialization of two matrices, so $O(1)$ complexity and $O(n^2)$ memory consumption are

required. Therefore, this part of the algorithm results in $O(max(E_c, n^2))$ complexity and $O(E_m, n^2)$ memory consumption.

---

**Algorithm 1** L-EXPM

---

1: **procedure** L-EXPM($M$)
2:     $\Lambda \leftarrow eigs(M)$
3:     $\Lambda \leftarrow controlEigs(\Lambda)$
4:     $r \leftarrow e^{\Lambda[1]}$
5:     $P \leftarrow I^{dim(M) \times dim(M)}$
6:     $i = 1$
7:     **while** $i \leq dim(M)$ **do**
8:         $expm \leftarrow expm + r \cdot P$
9:         $r \leftarrow e^{\Lambda[i]} \Sigma_{k=0}^{i} \Sigma_{l=0}^{i} \left( \frac{-1}{(\lambda_i - a_k)} \right)^{l+1} c_l e^{(a_k - \Lambda[i])} \Sigma_{m=0}^{l} \frac{l!}{(l-m)!} (\Lambda[i] - a_k)^{l-m}$
10:        $P \leftarrow P(M - I^{dim(M) \times dim(M)} \Lambda[i+1])$
11:     **end while**
12:     *return expm*
13: **end procedure**

---

Regarding lines 7–11, we repeat the inner loop $n$ times. Inside the inner loop at line 8, there is addition between two matrices, which has $O(n^2)$ complexity and memory consumption. In line 9, there are three sums, each bounded by $n$. In addition, in the inner sum, there is a factorial, which is naively calculated in $O(l)$ in each loop and can be bounded by $O(n)$ overall, and therefore, in the worst case, it can be calculated as $O(n^4)$ complexity and $O(n)$ memory consumption. Nevertheless, calculating the size of the factorial and storing it in a vector of size $n$ reduces the complexity of line 9 to $O(n^3)$. In line 10, the more expensive computation is the matrix multiplication, which is bounded by $O(n^3)$ complexity and $O(n^2)$ memory consumption. Therefore, the algorithm's complexity is bounded by $O(n^4)$ complexity and $O(n^3)$ memory consumption. The values of $a_k$ and $c_l$ are determined during the run time as shown in the Proof of Lemma 1.

*2.2. Proof of Soundness and Completeness*

In this section, we provide analytical proof for the soundness and completeness of the L-EXPM algorithm. The proof outlines how to reduce the formula proposed in L-EXPM, lines 8–19, to the formula used in Putzer's algorithm [26].

**Theorem 1.** *For any given matrix $M \in \mathbb{C}^{n \times n}$, algorithm L-EXPM solves Equation (1).*

**Proof.** Consider the following equation:

$$Y'(t) = MY(t), M \in \mathbb{C}^{n \times n}.$$

First, one needs to obtain the eigenvalues of the input matrix $M$. It is possible to find the biggest eigenvalue of $M$ using the Lanczos algorithm [27]. Using Ojalvo and Newman's algorithm, this process is numerically stable [28]. The result of their algorithm is a tridiagonal matrix $T_{m \times m}$, which is similar to the original matrix $M$. The eigenvalues of $T_{m \times m}$ can be obtained with an error $\epsilon$ as small as needed using spectral bisection [29].

According to the Putzer algorithm [26], a solution to the ME takes the form:

$$
\begin{aligned}
&Y(t) = \Sigma_{j=0}^{n-1} r_{j+1}(t) P_j, \text{ such that} \\
&P_0 = I, \ \forall j \in [0, n-1] : P_j = \Pi_{k=1}^{j} (M - \lambda_k I), \text{ and} \\
&r'_1 = \lambda_1 r_1, \ r_1(0) = 1, \ \forall j \in [2, n-1] : r'_j = \lambda_j r_j + r_{j-1}, r_j(0) = 0,
\end{aligned}
\tag{3}
$$

where $\lambda_i \in \Lambda$ are the eigenvalues of $M$ ordered from the absolute largest eigenvalue to the smallest one. The solution for $r_j' = \lambda_j r_j$ for any $j \in [1, n-1]$ is a case of a non-heterogeneous linear ODE with a constant coefficient and, therefore, takes the form:

$$r_1 = e^{\lambda_1 t}, \; r_j = e^{\lambda_j t}\left(\int e^{-\lambda_j t} r_{j-1}(t)dt - \int r_{j-1}(t)dt_{|t=0}\right) \tag{4}$$

Regarding $P_j$, it is possible to obtain $P_{j+1}$ given $P_j$ using the formula:

$$P_{j+1} = P_j(M - \lambda_{j+1}I), \tag{5}$$

because

$$P_{j+1} := \Pi_{k=1}^{j+1}(M - \lambda_k I) \text{ and } P_j := \Pi_{k=1}^{j}(M - \lambda_k I) \rightarrow P_{j+1} := P_j \Pi_{k=j+1}^{j+1}(M - \lambda_k I) = P_j(M - \lambda_{j+1}I).$$

Equation (1) may diverge if $\lambda_1 = 0$: then $r_j = \frac{(-1)^j}{\lambda_{j-1}\lambda_j}$ for $2 \leq j \leq n$ and $r_1 = 1$. In general, this calculation has only precision errors. It may have a significant cancellation error when $\lambda_{j-1}\lambda_j \rightarrow 0$.

Now, one needs to show that the solution for $r_j$ takes the form shown in line 9 of the L-EXPM algorithm.

**Lemma 1.** $\forall j \in [1, n] : r_j(t) := \Sigma_{k=1}^{n} p_k(t)e^{a_k t}$, where $p_k$ are polynomials and $a_k \in \mathbb{C}$ are constants.

**Proof of Lemma 1.** Induction on $j$. For $j = 1$, $r_1 = e^{\lambda_1 t} - 1$, which satisfies the condition. Assume the condition is satisfied for $r_j$, and we show $r_{j+1}$ has satisfied the condition. Now, according to Equation (4):

$$r_{j+1} = e^{\lambda_{j+1} t}\int e^{-\lambda_{j+1} t} r_j(t)dt - e^{\lambda_{j+1} t}\int r_j(t)dt_{|t=0}. \tag{6}$$

The second term is the multiplication of a constant by an exponent and therefore satisfies the condition. Now, we examine the first term. Based on the assumption, it takes the form:

$$r_{j+1} = e^{\lambda_{j+1} t}\int e^{-\lambda_{j+1} t}\Sigma_k p_k(t)e^{a_k t}dt = e^{\lambda_{j+1} t}\int \Sigma_k p_k(t)e^{(a_k - \lambda_{j+1})t}dt. \tag{7}$$

Integration of the sum is equal to the sum of the integrations, so we can write:

$$r_{j+1} = e^{\lambda_{j+1} t}\Sigma_k \int p_k(t)e^{(a_k - \lambda_{j+1})t}dt. \tag{8}$$

Therefore, it is enough to show that $\int p_k(t)e^{(a_k - \lambda_{j+1})t}dt$ satisfies the condition, because a sum of sums of elements that satisfy the condition also satisfies the condition. Now, $p_k(t)$ is a polynomial, and therefore it is possible to obtain:

$$r_{j+1} = \int (c_0 + c_1 t + \cdots + c_l t^l)e^{(a_k - \lambda_{j+1})t}dt = \int c_0 e^{(a_k - \lambda_{j+1})t} + \int c_1 t e^{(a_k - \lambda_{j+1})t} + \cdots + \int c_l t^l e^{(a_k - \lambda_{j+1})t}dt. \tag{9}$$

As a result, if one shows that $\int c_l t^l e^{(a_k - \lambda_{j+1})t}dt$ satisfies the condition, the whole term satisfies the condition. Now,

$$\int c_l t^l e^{(a_k - \lambda_{j+1})t}dt = \int_{(\lambda_{j+1} - a_k)t}^{\infty} \frac{c_l}{(\lambda_{j+1} - a_k)^l}x^l e^{-x}dx. \tag{10}$$

As this integral is the incomplete gamma function, it is possible to approximate it as follows:

$$\left(\frac{-1}{(\lambda_{j+1} - a_k)}\right)^{l+1} c_l e^{(a_k - \lambda_{j+1})t} \Sigma_{m=0}^l \frac{l!}{(l-m)!} ((\lambda_{j+1} - a_k)t)^{l-m}. \tag{11}$$

Equation (11) is an exponent of a constant and a sum of polynomials and therefore satisfies the condition. Therefore, the first term in Equation (6) satisfies the condition. Therefore, $r_j$ for $j \in [1, n-1]$ satisfies the condition. $\square$

Since the Putzer algorithm [26] solves Equation (1), it is enough to show that it is possible to obtain $r_j$ using line 9 because the L-EXPM algorithm is an approximation of the Putzer algorithm and therefore solves Equation (1).

Now, for $j = 1$, $r_1(t) = e^{\lambda_1 t} - 1$. According to Lemma 1, $r_2$ takes the form:

$$r_2(t) = e^{\lambda_2 t} \int e^{-\lambda_2 t} (e^{\lambda_1 t} - 1) dt - e^{\lambda_2 t} \int e^{\lambda_1 t} - 1 dt_{|t=0} = e^{(\lambda_2 + \lambda_1)t} (\frac{1}{\lambda_1 - \lambda_2} + \frac{1}{\lambda_2} + \alpha_2) - e^{\lambda_2 t}(\frac{1}{\lambda_1} - t + \alpha_1), \tag{12}$$

where $\{\alpha_i\}_{i=1}^2$ are constants. It is known that $r_2(0) = 1$, so by setting $t = 0$ in Equation (12), one obtains:

$$r_2(t) = (\frac{\lambda_1}{\lambda_2(\lambda_1 - \lambda_2)}) e^{(\lambda_2 + \lambda_1)t} - (\frac{1}{\lambda_1} - t) e^{\lambda_2 t} + \frac{\lambda_1^2 - \lambda_1\lambda_2 - \lambda_2}{\lambda_1(\lambda_1 - \lambda_2)}. \tag{13}$$

This process repeats itself for any $r_j$, $j \in [3, n-1]$. $\square$

## 3. Numerical Algorithm Evaluation

In comparison with other algorithms, L-EXPM's storage requirements and complexity are asymptotically equal to the ones of the algorithms with the lowest storage requirements and complexity [8]. To evaluate the performance of L-EXPM compared to the state-of-the-art ME algorithms, a general case matrix may be required. Nevertheless, since ME algorithms provide large errors if the given matrix has a specific property, several families of stiff and non-stiff matrices should be used to obtain a general case analysis of the algorithm's performance.

### 3.1. Artificial Stiff Matrices Analysis

We evaluate the performance of L-EXPM with respect to four state-of-the-art ME algorithms: Taylor, Pade, Newton, and Lagrange. For each algorithm, the parameters and tolerances are obtained using the grid search method [30]. Namely, for the Taylor algorithm, the number of terms is determined. In a similar manner, for the Pade algorithm, the fixed degrees combined with scaling and squaring [31,32] and the tolerance parameter are determined using the grid search method. The evaluation of the algorithms is performed using their MATLAB implementations (version 2020b). Since all five algorithms handle random, non-stiff matrices, we compared these algorithms on seven types of stiff matrices:

1.  Matrices for which the difference between the eigenvalues of the matrix is small but not negligible: we randomly pick a value ($a > 0$) and an amplitude ($\epsilon << 1$) and generate matrices with eigenvalues that are in the range ($a \pm \epsilon$).
2.  Matrices for which the eigenvalues are approaching 0: we generate matrices with eigenvalues that satisfy the following formula: $1 \le i \le n, \lambda_i = \frac{1}{(i+2)^2}$.
3.  Matrices with large diameters: we generate matrices with eigenvalues that satisfy the formula
    $0 \le i \le n, \lambda_i = a - \frac{(a-b)i}{n}$, where $a$ and $b$ are picked randomly such that $b >> a$.
4.  Matrices that have a large condition number: we generate matrices with eigenvalues that satisfy the formula $1 \le i \le n, \lambda_i \in [a, b]$, such that $\frac{|b|}{|a|} >> 1$.
5.  Matrices that have eigenvalues with significant algebraic multiplicity: we generate matrices with eigenvalues with an algebraic multiplicity of at least two.
6.  Matrices with a single eigenvalue: we generate matrices with a single eigenvalue picked at random.

7. Matrices with complex eigenvalues with a large imaginary part: we generate matrices with eigenvalues that satisfy the formula $0 \leq j \leq n, \lambda_i = a + 10i \cdot a$, where $a \in [-100, 100]$ is a random number.

The matrices are generated as follows. First, a Jordan matrix ($J$) with the required eigenvalues is randomly generated. Then, a random matrix $P$ with the same size of $J$ is generated such that the condition number of $P$ is less than two, and its determinant $0.5 \leq det(P) \leq 1.5$. The random matrix used in the analysis is obtained by computing $M = P^{-1}JP$. For each type of matrix, we examine the performance of each algorithm on matrices with sizes $3 \times 3, 10 \times 10, 100 \times 100$, and $1000 \times 1000$ to determine the growth of the error as a function of the matrix's size. Each value is obtained as the average of $n = 100$ repetitions, and the values of the matrices are generated using a normal distribution with mean $\mu = 0$ and standard deviation $\sigma = 300$.

The numerical relative error is calculated using the $L_2$ distance between the analytically obtained (ground truth) ME matrix and the numerically obtained one: $Error = ||A(M) - e_{gt}^M||_{L_2}$, where $A$ is a numerical ME algorithm. The ground truth ME is obtained analytically by calculating the ME of the $J$ matrix to obtain $e_{gt}^M = P^{-1}e^J P$. While $(e_{gt}^M$ is a closed form, it is not guaranteed to be exact since the matrix $(e_{gt}^M$ is a product of performing finite arithmetic. However, matrix $(e_{gt}^M$ is computed as an ME of a diagonal matrix, which is formed by computing exactly the exponential of a floating point number and is assumed to be an accurate computation (up to $\epsilon$-machine), and the multiplication with a random matrix with a small condition number (smaller than 2) is numerically stable [33]. Therefore, while the matrix $(e_{gt}^M$ is not guaranteed to be exact, it is close enough to the ground truth for any applied purpose. The results of this analysis are shown in Table 1.

**Table 1.** Numerical relative error from four numerical ME algorithms and the L-EXPM ME algorithm of seven stiff cases of matrices for matrix sizes $3 \times 3, 10 \times 10, 100 \times 100$, and $1000 \times 1000$. The results are the average of $n = 100$ random matrices in each case. Err indicates an error in the computation due to stack overflow.

| Matrix Size | Algorithm | Type 1 | Type 2 | Type 3 | Type 4 | Type 5 | Type 6 | Type 7 |
|---|---|---|---|---|---|---|---|---|
| $3 \times 3$ | Naive | $2 \times 10^{-3}$ | $3 \times 10^{-3}$ | $6 \times 10^{-2}$ | $8 \times 10^{-1}$ | $3 \times 10^{-3}$ | $8 \times 10^{-4}$ | $1 \times 10^{-3}$ |
| $3 \times 3$ | Pade | $\mathbf{5 \times 10^{-5}}$ | $6 \times 10^{-5}$ | $\mathbf{6 \times 10^{-4}}$ | $9 \times 10^{-3}$ | $2 \times 10^{-5}$ | $7 \times 10^{-5}$ | $1 \times 10^{-4}$ |
| $3 \times 3$ | Newton | $8 \times 10^{-3}$ | $5 \times 10^{-2}$ | $3 \times 10^{-2}$ | $9 \times 10^{-2}$ | $5 \times 10^{-5}$ | $\mathbf{9 \times 10^{-7}}$ | $7 \times 10^{-6}$ |
| $3 \times 3$ | Lagrange | $3 \times 10^{-2}$ | $1 \times 10^{-2}$ | $4 \times 10^{-2}$ | $8 \times 10^{-2}$ | $1 \times 10^{-6}$ | $1 \times 10^{-6}$ | $\mathbf{8 \times 10^{-7}}$ |
| $3 \times 3$ | L-EXPM | $1 \times 10^{-4}$ | $\mathbf{8 \times 10^{-6}}$ | $8 \times 10^{-4}$ | $\mathbf{8 \times 10^{-3}}$ | $\mathbf{9 \times 10^{-7}}$ | $1 \times 10^{-6}$ | $3 \times 10^{-7}$ |
| $10 \times 10$ | Naive | $4 \times 10^{-1}$ | $3 \times 10^{-1}$ | $1 \times 10^{0}$ | $8 \times 10^{0}$ | $4 \times 10^{-1}$ | $8 \times 10^{-1}$ | $1 \times 10^{0}$ |
| $10 \times 10$ | Pade | $\mathbf{1 \times 10^{-4}}$ | $5 \times 10^{-5}$ | $2 \times 10^{-3}$ | $2 \times 10^{-1}$ | $8 \times 10^{-2}$ | $8 \times 10^{-2}$ | $1 \times 10^{-2}$ |
| $10 \times 10$ | Newton | $2 \times 10^{-2}$ | $5 \times 10^{-1}$ | $8 \times 10^{-1}$ | $3 \times 10^{0}$ | $5 \times 10^{1}$ | $\mathbf{1 \times 10^{-3}}$ | $3 \times 10^{-3}$ |
| $10 \times 10$ | Lagrange | $2 \times 10^{-2}$ | $3 \times 10^{-1}$ | $1 \times 10^{0}$ | $4 \times 10^{0}$ | $7 \times 10^{1}$ | $5 \times 10^{-3}$ | $\mathbf{2 \times 10^{-3}}$ |
| $10 \times 10$ | L-EXPM | $4 \times 10^{-4}$ | $\mathbf{1 \times 10^{-6}}$ | $2 \times 10^{-3}$ | $\mathbf{8 \times 10^{-2}}$ | $\mathbf{6 \times 10^{-2}}$ | $7 \times 10^{-2}$ | $5 \times 10^{-3}$ |
| $100 \times 100$ | Naive | $2 \times 10^{0}$ | $8 \times 10^{-1}$ | $4 \times 10^{1}$ | $6 \times 10^{2}$ | $3 \times 10^{1}$ | $2 \times 10^{1}$ | $8 \times 10^{1}$ |
| $100 \times 100$ | Pade | $7 \times 10^{-2}$ | $5 \times 10^{-4}$ | $\mathbf{9 \times 10^{-1}}$ | $\mathbf{3 \times 10^{0}}$ | $\mathbf{2 \times 10^{0}}$ | $1 \times 10^{1}$ | $5 \times 10^{0}$ |
| $100 \times 100$ | Newton | $4 \times 10^{0}$ | $1 \times 10^{3}$ | $5 \times 10^{1}$ | $5 \times 10^{1}$ | $3 \times 10^{2}$ | $\mathbf{2 \times 10^{0}}$ | $7 \times 10^{0}$ |
| $100 \times 100$ | Lagrange | $3 \times 10^{0}$ | $1 \times 10^{3}$ | $7 \times 10^{1}$ | $1 \times 10^{2}$ | $8 \times 10^{2}$ | $\mathbf{2 \times 10^{0}}$ | $8 \times 10^{0}$ |
| $100 \times 100$ | L-EXPM | $\mathbf{3 \times 10^{-2}}$ | $\mathbf{3 \times 10^{-4}}$ | $1 \times 10^{0}$ | $5 \times 10^{0}$ | $8 \times 10^{0}$ | $4 \times 10^{0}$ | $1 \times 10^{1}$ |
| $1000 \times 1000$ | Naive | $2 \times 10^{5}$ | $7 \times 10^{6}$ | Err | Err | Err | $7 \times 10^{6}$ | Err |
| $1000 \times 1000$ | Pade | $8 \times 10^{2}$ | $\mathbf{2 \times 10^{3}}$ | Err | Err | $\mathbf{3 \times 10^{3}}$ | $6 \times 10^{2}$ | $\mathbf{1 \times 10^{4}}$ |
| $1000 \times 1000$ | Newton | $3 \times 10^{4}$ | Err | Err | Err | Err | $1 \times 10^{3}$ | $3 \times 10^{4}$ |
| $1000 \times 1000$ | Lagrange | $2 \times 10^{4}$ | Err | Err | Err | Err | $3 \times 10^{3}$ | $6 \times 10^{4}$ |
| $1000 \times 1000$ | L-EXPM | $\mathbf{4 \times 10^{1}}$ | $4 \times 10^{5}$ | Err | Err | $5 \times 10^{3}$ | $\mathbf{7 \times 10^{1}}$ | $2 \times 10^{4}$ |

Based on the results shown in Table 1, we compare the average relative error across the seven types of matrices of each algorithm divided by the matrix sizes, as shown in Figure 2, where each point is the mean value of each row in Table 1. The $x$-axis is the

matrix's size (i.e., dimension), and the y-axis is the numerical error as computed by the $L_2$ norm metric between the analytically and numerically obtained matrices. Unsurprisingly, all five algorithms present monotonically increasing numerical error with respect to the input's matrix size.

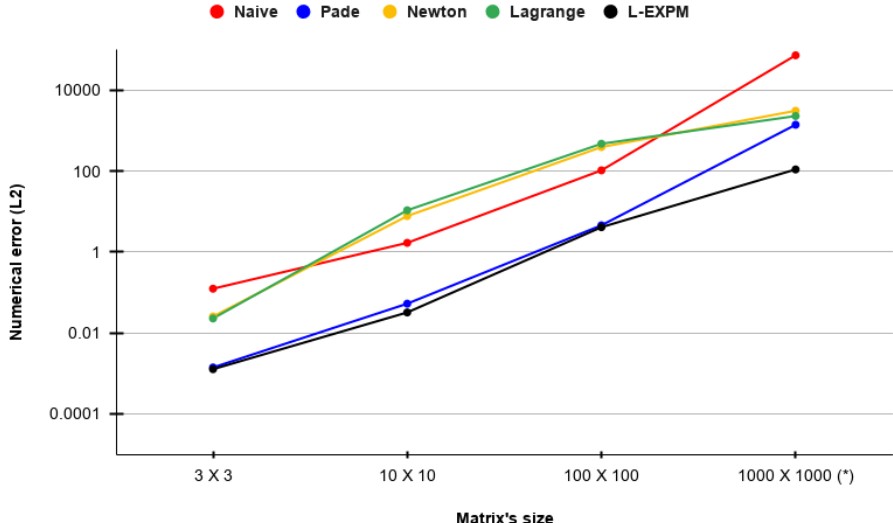

**Figure 2.** Average relative numerical error of the five ME algorithms across the five types of matrices divided by matrix size. * The values for the ($1000 \times 1000$) case are the average of the matrices from types 1 and 6 (see Table 1) rather than all seven types.

### 3.2. Control System's Observability Use Case

One usage of MEs is linear control systems, wherein one records the output of a control system over time and wishes to obtain the input of the control system, which is known as the *observability* problem [34]. Formally, one observes the output from the system:

$$\frac{dx}{dt} = Ax, \; y = Cx, \tag{14}$$

where $A, C \in \mathbb{R}^{n \times n}$ over a finite period of time $[t_0, T]$, and we aim to compute $x(t_0)$. Analytically, this problem is solved by [35] and requires computing the ME of $A$ and $C$ (see [34] for more details). Based on the glucagon–glucose dynamics linear control system for the regulation of artificial pancreas solution in type 1 diabetes [36], we simulated $A$ and $C$ matrices of sizes $3 \times 3$, $11 \times 11$, $100 \times 100$, and $1000 \times 1000$ that follow the same distribution. In particular, we introduce Gaussian noise with a mean of 0.01 and standard deviation of 0.001 for all non-zero values in the matrix to simulate measurement error. This way, one obtains realistic samples of matrices that one can find in clinical settings. We used a dedicated server with an Ubuntu 18.04.5 operating system. The server had an Intel Core i7-9700K CPU and 64 GB RAM . All experiments were conducted in a linear fashion, and no other programs were executed on the device except for the operating system. This was to ensure the computation time was measured accurately. Each matrix's size was computed with 100 different samples obtained by using the different seeds for the pseudo-random process. The results are shown as mean $\pm$ standard deviation in Table 2.

Taken jointly, the resultsshow that L-EXPM has similar or slightly better numerical stability compared to current state-of-the-art ME algorithms for stiff matrices. For the more general case, L-EXPMs show statistically significant ($p < 0.05$ with paired two-tailed $t$-test) better error compared to the Pade algorithm. This outcome comes with a cost of one or two orders of magnitude more computational time.

**Table 2.** Comparison between the Pade approximation algorithm's and the proposed L-EXPM algorithm's errors and computation times (seconds) for the observability task.

| Algorithm | Metric | $3 \times 3$ | $10 \times 10$ | $100 \times 100$ | $1000 \times 1000$ |
|---|---|---|---|---|---|
| Pade | Error | $5.2 \times 10^{-3} \pm 2.8 \times 10^{-4}$ | $9.8 \times 10^{-3} \pm 1.0 \times 10^{-3}$ | $6.5 \times 10^{-1} \pm 8.3 \times 10^{-2}$ | $8.3 \times 10^{0} \pm 5.1 \times 10^{-1}$ |
| | Time | $3.1 \times 10^{-3} \pm 0.2 \times 10^{-3}$ | $7.0 \times 10^{-1} \pm 0.4 \times 10^{-1}$ | $5.8 \times 10^{1} \pm 5.9 \times 10^{0}$ | $3.2 \times 10^{4} \pm 0.7 \times 10^{4}$ |
| L-EXPM | Error | $4.3 \times 10^{-3} \pm 7.5 \times 10^{-5}$ | $6.9 \times 10^{-3} \pm 7.7 \times 10^{-4}$ | $2.1 \times 10^{-2} \pm 4.1 \times 10^{-3}$ | $7.6 \times 10^{-1} \pm 1.3 \times 10^{-1}$ |
| | Time | $5.2 \times 10^{-1} \pm 0.9 \times 10^{-1}$ | $1.4 \times 10^{1} \pm 0.2 \times 10^{1}$ | $2.2 \times 10^{2} \pm 0.3 \times 10^{2}$ | $8.5 \times 10^{5} \pm 1.1 \times 10^{5}$ |

## 4. Matrix Exponential Decision Tree

As shown in Table 1, there is no one ME algorithm that "rules them all" and outperforms all other ME algorithms for all cases in terms of numerical error. On top of that, we have neglected the computation time and resources needed to perform these algorithms on different matrices. Therefore, we can take advantage of the decision tree (DT) model. DTs are one of the most popular and efficient techniques in data mining and have been widely used thanks to their relatively easy interpretation and efficient computation time [37,38]. A DT is a mathematical tree graph wherein the root node has all the data, non-leaf nodes operate as decision nodes, and the leaf nodes store the model's output for a given input that reaches them.

Two types of DTs are important for computational systems: For one, the numerical error is critical to the result such that the resources and computing time are less significant. The second case is where the numerical error is less significant, while the time and resources required to obtain the results need to be minimized.

To find these DTs, we first generate a data set ($\mathbb{M}$) with 1000 matrices with sizes ranging between 3 and 100 generated for each of the first seven groups and an additional 7000 matrices from the eighth group generated to balance between stiff and non-stiff matrices (14,000 matrices in total). Afterward, the data set is divided into a *training* cohort and a *testing* cohort such that 80% of each sub-group of the data set is allocated to the training cohort, and the remaining 20% is allocated to the testing cohort. This process is repeated five times to obtain a five-fold split [39].

We aim to find a DT that is both computationally optimized and with minimal error and, as such, offers an optimal decision algorithm for the ME algorithm for any case while simultaneously taking into consideration computation time and numerical error. Therefore, we first define the available leaves and decision nodes for the DT. Each leaf node is a numerical ME algorithm with its own complexity, and each decision node is a discrete classification function that receives a matrix $M$ and returns a value $v \in \mathbb{N}$. The components used to construct the DT are shown in Table 3.

We define a penalty function as the sum of the worst-case complexity of all the computational components (vertices, marked by $v \in V$) computed during the DT. In addition, to avoid over-fitting the optimization on a too-small DT that does not use anything and uses the algorithm with the least worst-case complexity, a decrease by one order of magnitude (factor of 10) in the relative error is equal to dividing by a linear factor from the worst overall complexity. Formally, one can write the optimization problem as follows:

$$\min_{DT} \left( \Sigma_{M \in \mathbb{M}} \frac{\Sigma_{v \in V} C[v]}{log_{10} E} \right), \tag{15}$$

where $\mathbb{M}$ is the training set of matrices. We search for a directed acyclic graph (DAG) ($G = (V, E)$) for which for any pair of nodes $i \neq j \in \mathbb{N} : v_i, v_j \in V$ for which there is a path from $v_i$ to $v_j$ satisfying that the asymptotic complexity of $v_i$ is smaller or equal to $v_j$.

In addition, for the minimal error constraint, we compute the outcome for each one of the decision components shown in Table 3. We then compute the numerical relative error of each of the leaf components shown in Table 3 and store the index of the algorithm.

**Table 3.** The components that are available for the complexity-optimized DT.

| Index | Component | Description | Worst Case Complexity ($C$) | Average Relative Numerical Error ($E$) | Is Leaf Node |
|---|---|---|---|---|---|
| 1 | Check form | The matrix is diagonal, trigonal, Jordan, or full | $O(n^2)$ | 0 | No |
| 2 | Is symmetric | The matrix is symmetric or not | $O(n^2)$ | 0 | No |
| 3 | Large diameter | The diameter of the matrix is larger than some threshold or not | $O(n^3)$ | 0 | No |
| 4 | Large algebraic multiplicity | There are eigenvalues with algebraic multiplicity that are larger than some threshold or not | $O(n^3)$ | 0 | No |
| 5 | Large condition number | The condition number of the matrix is larger than some threshold $x$ or not | $O(n^3)$ | 0 | No |
| 6 | Single eigenvalue | Does the matrix have a single eigenvalue | $O(n^3)$ | 0 | No |
| 7 | Complex eigenvalues | Eigenvalues are complex with a big imaginary part | $O(n^3)$ | 0 | No |
| 8 | Close eigenvalues | Eigenvalues $\lambda_i, \lambda_j$ such that $|\lambda_i - \lambda_j| < const \land \lambda_i \neq \lambda_j$ | $O(n^3)$ | 0 | No |
| 9 | Diagonal | Diagonal matrix exponential | $O(n)$ | 0.13 | Yes |
| 10 | Jordan | Jordan matrix exponential | $O(n^4)$ | 21.21 | Yes |
| 11 | Eigenvector algorithms | TRED2 [40] and TQL2 [8] | $O(n^4)$ | 18.89 | Yes |
| 12 | Ill condition algorithms | IMPSUB [8] | $O(n^4)$ | 47.05 | Yes |
| 13 | L-EXPM | The proposed L-EXPM algorithm | $O(n^4)$ | 8.45 | Yes |
| 14 | Eigenvalue-based approximation | Lagrange algorithm [8] | $O(n^5)$ | 54.98 | Yes |
| 15 | Different eigenvalue approximation | Newton algorithm [8] | $O(n^5)$ | 52.73 | Yes |
| 16 | Power series approximation | Pade approximation algorithm [8] | $O(n^4)$ | 11.09 | Yes |
| 17 | Naive | Naive algorithm | $O(n^4)$ | 78.20 | Yes |

Based on both the complexity and numerical error data sets, a genetic programming approach has been used to obtain the DT model [41,42]. First, an initial population of DT models is generated as follows: based on the generated numerical related data set, a DT model is trained using the *CART* algorithm and the *gini* dividing metric [43]. In addition, the grid search method [30] is used on the DT's depth (ranging between two and seven levels) to obtain the best depth of the tree. Finally, the Boolean satisfiability-based post-pruning (SAT-PP) algorithm is used to obtain the smallest DT with the same level of accuracy [44]. The population of the DT model differs in two parameters: first, the maximum number of leaves, if the leaf node can be used twice or not, and the minimum samples for dividing a node [30].

Afterward, in each algorithmic step, each DT model is scored based on Equation (15) (without the optimization term)—performed as the *fitness* metric. The scores of all models are normalized such that the sum of the values equals 1. The top $p$ percent ($p$ is empirically picked to be 50%) of the population is kept for the next generation. The population is repopulated based on stochastic *mutations* of the remaining models. The mutation function operates as follows: First, two DT models are picked at random with a distribution corresponding to the fitness score of the models. Both DTs are scanned from the root node using the BFS algorithm [45] such that each node that they have is similarly allocated to the

new DT model, and nodes that are different are taken from the first model 33% of the time and from the second model 33% of the time, and the remaining 34% are pruned.

The obtained DT model is shown in Figure 3, wherein each rectangle node is a decision node and each circle node is a leaf node, which is identified by its component name as defined in Table 3.

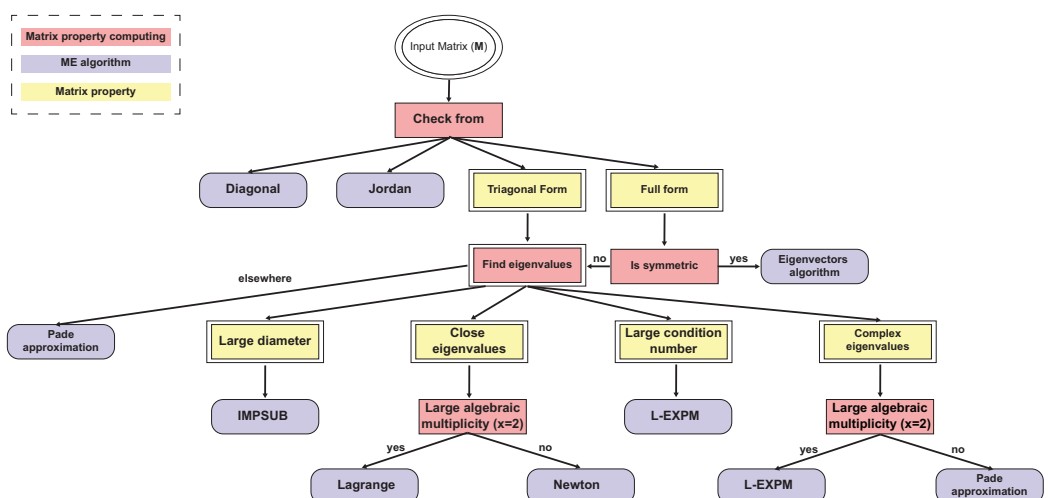

**Figure 3.** Numerical-accuracy- and computation-complexity-optimized DT for numerical MEs.

## 5. Conclusions

This study introduces a novel algorithm for numerically calculating matrix exponentials (MEs) that achieves high accuracy on stiff matrices while maintaining computational and memory efficiency comparable to existing ME algorithms. By combining eigenvalue-based and series-based approaches, the proposed L-EXPM algorithm demonstrates improved robustness against stiff matrices compared to individual methods. Although L-EXPM generally outperforms the Pade approximation algorithm, especially for large matrices, the latter remains preferable for time-sensitive applications due to its shorter computation time. In practical applications, like observability in control systems, L-EXPM shows superior accuracy but with longer computation times than Pade. To address varying matrix characteristics, a decision tree (DT) model integrating Boolean functions and ME algorithms is proposed, and it is optimized using machine learning and genetic programming techniques.

This study is not without limitations. First, an analytical boundary for the error, rather than the numerical one shown in this study, can be theoretically useful. Second, exploring the influence of different eigenvalue computation methods can further improve the numerical performance of L-EXPM. Finally, future research should focus on testing this DT model in diverse engineering contexts, particularly in control systems, to assess its performance across real-world scenarios.

**Author Contributions:** T.L.: conceptualization, data curation, methodology, formal analysis, investigation, supervision, software, visualization, project administration, and writing—original draft. S.B.-M.: validation and writing—review and editing. All authors have read and agreed to the published version of the manuscript.

**Funding:** This research did not receive any specific grant from funding agencies in the public, commercial, or not-for-profit sectors.

**Data Availability Statement:** All the data that were used were computed.

**Conflicts of Interest:** The authors have no conflicts of interest to declare that are relevant to the content of this article.

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
