# Peer review of "More Numerically Accurate Algorithm for Stiff Matrix Exponential"

_mathematics, doi:10.3390/math12081151_

Round 1
Reviewer 1 Report
Comments and Suggestions for Authors
Needs minor editing
Author Response
First of all, we would like to thank the editor and reviewers for their positive and professional review. The comments allowed us to improve the quality of the work and make it more suitable for high-quality journal like Mathematics.
Comment: “Term or terms should be defined first before abbreviating.”
Answer: Thank you for pointing it out, following this comment, we make sure each abbreviation is provided at its first appearance. DT - decision tree, ME - matrix exponent, ODE - ordinary differential equation.
Comment: “The second sentence after equation (1) needs modification”
Answer: Thank you for alerting us about this point - Fixed. In a more general sense, we carefully edited the paragraph before and after to make sure the text reads better.
Comment: “The parameters in the equations must be defined”
Answer: Thank you for this comment - fixed.
Comment: “Some words are itallised”
Answer: Thank you for this observation. Indeed, we used italic to emphasize some words are names, following the following guideline: https://www-s3-live.kent.edu/s3fs-root/s3fs-public/file/italics.pdf?VersionId=goWX1GDBFsYs4A29sHlAy1JYMUlsam7h#:~:text=Italics%20are%20used%20primarily%20to,in%20writing%2C%20but%20only%20rarely.
We did not see any reference to this point in the journal’s style guideline so we hope this is OK.
Comment: “There is need for improvement in the language. For instance, “if for some reason should be if for some reasons”.”
Answer: Thank you very much for this suggestion. First of all, we took this point very seriously and used professional proofreading services with native English-speaking individuals in order to make sure the paper was written in a correct and approachable manner.
Comment: “An equation is not numbered. Check section 2.2”
Answer: Thank you for alerting us about this stylistic shortcoming. We added an equation number for the missing place in the first equation in section 2.2.
Comment: “Correct abnormalities in equation (3)”
Answer: Thank you for this suggestion. We divided the equation into its three parts with the connecting words in between.
Comment: “The two equations labeled in equation (4) should be on different lines”
Answer: Thank you - fixed.
Comment: “The title of table should be on top of the table and not below the table.”
Answer: Thank you for alerting us about this stylistic issue. Following this comment, we fixed it.
Comment: “Many paragraphs in the conclusion”
Answer: We alter the conclusion section completely, making it much shorter and with a single paragraph while preserving the main claims and overall structure and flow of this section.
Comment: “The references are old. There is just one 2020 and one 2022 references out of 39 references. This shows that there is not valuable in the concept”
Answer: Thank you for alerting us about this issue. Following this comment, we added multiple new studies from 2020 and after. In fact, one of them was published just half a year ago in this journal and uses ME, highlighting the fact this is an active field of research.
Comment: “Authors and their affiliations should be arranged properly.”
Answer: Thank you for alerting us about this stylistic issue. Following this comment, we fixed it.
Reviewer 2 Report
Comments and Suggestions for Authors
See the File

Author Response
First of all, we would like to thank the editor and reviewers for this professional feedback and for letting us revise the manuscript.
Comment: If we summarize the claim of the reviewer, the fact we did not analytically bound the error of the numerical error in the algorithm is not sufficient.
Answer: We would like to thank the reviewer for the valuable feedback. Indeed, we were not able to provide a proof for the algorithm’s stability. We made another attempt following this comment but without any luck. As this is indeed a limitation of the proposed work, we numerically explored the algorithm with multiple cases and compared it to the state of the art. As shown in the obtained results, the algorithm outperforms other algorithms for stiff cases, regardless of the criticism raised by the reviewer. As such, we believe that despite the limitations the reviewer pointed out, the proposed work contributes to the field of numerical analysis as we are the first to show a numerical version of Putzer’s algorithm.
Round 2
Reviewer 1 Report
Comments and Suggestions for Authors
The authors need to highlight the corretions in the text for easy identification
Author Response
Comment: “The authors need to highlight the corretions in the text for easy identification.”
Answer: We apologize for the confusion. However, it seems that the changes are highlighted in bold text - we made sure this revision the changes from both rounds are presented clearly in bold.
Reviewer 2 Report
Comments and Suggestions for Authors
I think that Authors need to add some descriptions about stability of the algorithm and it’s current state. Also, the explanations of problems and difficulties for this question are useful. Possible ways for development of the algorithm need to be pointed out.
Author Response
Comment: “I think that Authors need to add some descriptions about stability of the algorithm and it’s current state. Also, the explanations of problems and difficulties for this question are useful. Possible ways for development of the algorithm need to be pointed out.”
Answer: Thank you very much for this suggestion. Following this comment, we performed several improvements to the paper. First, we further explain the stability of the algorithm based on the obtained results in the Results section. Second, we added to the Discussion several ideas on how one can use the algorithm and the DT in realistic systems.
------
Please note that the changes from both review rounds were kept in bold font for the convenience of the review.